# Effectiveness of Multicomponent Interventions and Physical Activity in the Workplace to Reduce Obesity: A Systematic Review and Meta-Analysis

**DOI:** 10.3390/healthcare11081160

**Published:** 2023-04-18

**Authors:** M. Rocío Jiménez-Mérida, Manuel Vaquero-Abellán, José M. Alcaide-Leyva, Vanesa Cantón-Habas, Elena Raya-Cano, Manuel Romero-Saldaña

**Affiliations:** 1Departamento de Enfermería, Farmacología y Fisioterapia, Facultad de Medicina y Enfermería, Universidad de Córdoba, 14014 Córdoba, Spain; n02jimem@uco.es (M.R.J.-M.);; 2Grupo Asociado de Investigación GA16 Estilos de Vida, Tecnología y Salud, Instituto Maimónides de Investigación Biomédica de Córdoba (IMIBIC), Departamento de Enfermería, Farmacología y Fisioterapia, Facultad de Medicina y Enfermería, Universidad de Córdoba, 14014 Córdoba, Spain

**Keywords:** health promotion, obesity, overweight, physical activity, workplace

## Abstract

Background: Overweight and obesity are public health problems that affects the workplace. This paper aims to analyse the effectiveness of workplace health promotion interventions in reducing Body Mass Index (BMI); Methods: Following PRISMA guidelines, a systematic review was conducted using PubMed, MEDLINE, and SCOPUS databases. The inverse variance statistical method was used for the meta-analysis with a random effects analysis model and standardised means. The results have been represented by Forest Plots and Funnel Plots graphs; Results: The multicomponent approach had the best results for reducing BMI (−0.14 [−0.24, −0.03], 95% CI; *p* = 0.009) compared to performing physical activity only (−0.09 [−0.39, 0.21], 95% CI; *p* = 0.56). However, both methods resulted in positive changes in reducing BMI in the general analysis (−0.12 [−0.22, −0.02], 95% CI; *p* = 0.01). The GRADE evaluation showed low certainty due to the high heterogeneity between interventions (I^2^ = 59% for overall analysis). Conclusions: The multicomponent approach could be an effective intervention to reduce obesity in the working population. However, workplace health promotion programs must be standardised to conduct quality analyses and highlight their importance to workers’ well-being.

## 1. Introduction

Overweight and obesity are public health problems that, according to the World Health Organization (WHO), in 2016, between 39% and 13% of the adult world population over 18 years of age suffer from [1]. In 2019, the Organization for Economic Cooperation and Development (OECD) released a report called “The Heavy Burden of Obesity”, which highlighted how obesity has increased in recent decades. Among the reasons for this increase are bad dietary habits, lack of physical activity, and a sedentary lifestyle, making obesity an epidemic of the 21st century. In addition, it estimates that in the next 30 years, this will cause around 220 million non-communicable diseases, including cardiovascular problems, diabetes, and a reduced life expectancy in people with cancer [2].

The workplace setting is one of the most affected areas by this issue. People with chronic illnesses are exposed to greater risk of missing more days from work and lower productivity. In addition, workers who suffer from overweight and obesity cause around 54 million sick leaves per year: 28 million corresponded to a reduction in employability, 18 million to reduced productivity, and the remaining 8 million to work absenteeism [2]. To all this, we must add that the consequences of the COVID-19 pandemic, which have worsened the already alarming figures in some sectors of the working population [3], particularly in those who work from home [4].

To classify overweight and obesity, the most widely used method is the Body Mass Index (BMI) as a reference measure. Although other parameters for measuring body fat allow a deeper and more individualized assessment [5], the BMI can be useful for a first classification [6,7].

Due to the implications of this problem on workers’ health and the repercussions at the socioeconomic level, numerous studies have attempted to address it using a workplace health promotion (WHP) approach. The most prevalent intervention is promoting physical activity (PA). In this sense, we found studies that have observed less work absenteeism among workers who performed PA at a moderate-vigorous level, at least three sessions a week, compared to those who did not [8,9], as well as an improvement in well-being [10]. These interventions have also shown good cost-effective results [11] and increased health [12]. On the other hand, nutritional educational programs are effective in helping workers choose healthy foods [13] and increasing the consumption of fruits and vegetables [14]. However, the problem is much more complex, and it is not only necessary to increase the level of PA and nutrition knowledge among workers. Multilevel or multicomponent interventions, which address healthy lifestyle habits and workers’ mental health and wellbeing, are presented as favourable options, being the most effective in addressing complex behavioural changes [15,16,17], such as the approach to overweight and obesity [18].

Despite multiple studies of occupational health promotion interventions aiming to improve workers’ health and reducing BMI value, more homogeneity in their implementation is needed. Heterogeneous interventions lead to results that cannot be adequately analysed and implemented in a general population, making it difficult for companies and specialized personnel to choose an effective WHP program. Further study and analysis of interventions and their results are needed to provide practical recommendations [10,19]. For this reason, this paper aims to analysse the effectiveness of workplace health promotion interventions in reducing BMI.

## 2. Materials and Methods

### 2.1. Search Methods and Strategy

In the present study, a systematic review was conducted using the PubMed, MEDLINE and SCOPUS databases. The search was done in September 2022. The PRISMA guide [20] was followed to structure and wrote the article.

The search strategy was the following: (“Health Promotion” OR “Total worker health” OR “Risk Reduction Behaviour”) AND (“Workplace”) AND (“obesity” OR “BMI” OR “overweight”)

Time filters (January 2015–June 2022) and types of study (clinical trials) were applied.

Two groups of researchers reviewed the results, selecting those oriented towards health promotion interventions at work.

### 2.2. Selection Criteria

Inclusion criteria:-Articles that include health promotion activities in companies aimed at reducing BMI.-Articles in English and Spanish.

Exclusion criteria:-Articles on health promotion interventions with populations outside the workplace.-Articles of health promotion interventions that did not carry out a post-intervention follow-up of the BMI value.

### 2.3. Selection of Studies

The first phase of the review was done by two groups of two researchers, who created a database with those articles selected from reading the title and abstract. Once finished, both databases were pooled to eliminate duplicate reports. A third team was in charge of reviewing the articles whose inclusion could be doubtful.

After preparing the database of general health promotion interventions in workplaces, those that met the study’s selection criteria were extracted. All reviewers had access to all data from the reviewed studies.

### 2.4. Data Collection Process

Independent teams performed data extraction and verification. The information analysed included the identification data (author, year, and country of the study/company), the characteristics of the participants (age and individual characteristics), the sample size, the study design, the description of the intervention, and the result of it.

### 2.5. Interventions and Population

The studies that have been included analyse the efficacy of specific WHP interventions on the population of active working age (18 to 65 years). Both individual and collective WHP interventions have been included, which measure the reduction of the BMI value.

### 2.6. Types of Studies and Outcome Variables

Randomised clinical trials were reviewed. The outcome variable was the BMI value after the proposed intervention.

### 2.7. Assessment of Risk of Bias in Individual Studies

The Cochrane Collaboration tool was used to assess the risk of bias through the RevMan version 5.4.1 program [21]. In this tool, the risk of individual bias was evaluated based on the following items: sequence generation (randomisation); allocation concealment; blinding of participants, personnel, and investigator; incomplete data; selective outcome reporting; and other possible sources of bias.

### 2.8. Quantitative Analysis

The meta-analysis was performed using the RevMan 5.4.1 software tool. The inverse variance statistical method was used with a random effects analysis model (randomisation with a 95% confidence interval (CI)) and standardised means. The results have been represented by Forest Plots and Funnel Plots graphs. Study heterogeneity was assessed using the I^2^ (inconsistency index), classifying heterogeneity as low (0–25%), moderate (26–50%), and high (51–100%). The Prediction Intervals (PI) for the effect size have been calculated using the method proposed by Borenstein [22]. The Grade Pro program [23] was used to summarise the finding and analyse the evidence, which assigns a value according to the certainty obtained (high, medium, low, or very low). The inconsistency of the evidence would be measured by the results of the inconsistency index analysis. The publication bias would be measured by the results shown in the funnel plot graph.

## 3. Results

Of the 385 studies, 12 were finally selected for the meta-analysis (Figure 1). Table 1 shows the main characteristics of each intervention. The origin of each program is diverse; the most numerous were those from the USA (4), followed by India (1), China (1), Malaysia (1), Canada (1), New Zealand (1), Iran (1), The Netherlands (1), and Brazil (1).

After the risk of bias analysis (Figure 2 and Figure 3), the item that obtained a score of less than 50% was the blinding of participants and personnel. This has been due to the difficulty of having complete blinding in the interventions since, in some cases, they were carried out in the same company and, in other cases, it was the research staff performing the intervention. The items allocation concealment and blinding of outcome assessment have obtained a low score, although in summary were higher than 50%. The risk of bias assessment summary has been low or medium-low, not excluding any of the selected articles for this reason.

Among the interventions to reduce BMI, we find multicomponent training interventions in healthy lifestyles [24,25,26,27,28,29,30,31] and single physical activity interventions [32,33,34,35]. To analyse the interventions, due to high heterogeneity, the studies were divided into two subgroups: interventions that implemented only physical exercise programs of at least 15 min/per day every day, or 30 min/per day at least five days a week, regardless of whether they were performed during the working day or during their free time; and multicomponent interventions that combined two or more different approaches.

**Table 1 healthcare-11-01160-t001:** Characteristics of the studies.

Author	Year	Country	N ^1^	Intervention	BMI and Other Results
Almeida et al. [28]	2015	USA	1499	**Multicomponent** Comparison of two programmes: Physical activity intervention and financial incentives (INCENT); lower intensity intervention (Living my weight (LMW)).	Participants lost an average of 2.27 lbs and 1.3 lbs in the INCENT and LMW interventions. Differences between the two programmes in weight loss and BMI reduction were not statistically significant.
Fernández et al. [26]	2015	USA	859	**Multicomponent** An intervention promoting healthy lifestyles through portion control, healthy diets, and PA to prevent weight gain in workers.	There was a decrease in the intervention groups in the number of overweight/obese workers by 3.7% compared to an increase of 4.9% in the control groups. Although these were not significant changes, there was a moderate improvement in the intervention participants.
Gu et al. [32]	2020	China	262	**Physical Activity** Group intervention to promote physical activity.	The intervention significantly improved physical activity level, walking time, systolic blood pressure, waist circumference, body fat percentage and BMI.
Jamal et al. [30]	2016	Malaysia	194	**Multicomponent** Eating behaviour modification programme through a supportive group intervention to modify lifestyle habits.	The intervention group lost 6% weight compared to 4.1% of the control group. After 24 weeks, 83.5% maintained the changes in their routine. In addition, there was an improvement in negative feelings, physical discomfort, perception of social support, and quality of life in this group.
MacEwen et al. [27]	2017	Canada	25	**Multicomponent** Intervention for workers with abdominal obesity: “Sit-Stand Desks”. Physical activity and organisational approach.	The intervention group had a significant reduction in daily and total sitting time and a daily increase in standing. There were no changes in cardiovascular risk markers.
Mansi et al. [33]	2015	New Zealand	58	**Physical Activity** Intervention to prevent sedentary lifestyles, monitoring the level of physical activity together with an educational programme.	The intervention group increased their average daily steps from 5993 to 9792, while the control group increased from 5788 to 65,551. This improvement was maintained after 12 weeks, along with the level of self-reported physical activity. There were no significant changes in the mental health component between groups.
Noori et al. [31]	2021	Iran	80	**Multicomponent** Empowerment programme to improve health habits.	There was a significant increase in the intervention group compared to the control group in the variables of nutrition, physical activity, stress management, interpersonal relationships, and health responsibility as measured by the Health Promoting Lifestyle Profile-II questionnaire.
Reif et al. [25]	2020	USA	4834	**Multicomponent** Intervention to promote healthy habits in the company with an integrated approach: wellness activities, financial incentives, and paid time off.	After the intervention, there was a significant improvement in health beliefs and behaviours, especially in decreasing body weight, blood pressure, cholesterol, and glucose levels; there were no significant changes in new medical diagnoses or the number of doctor visits after 24 months.
Renaud et al. [24]	2020	Netherlands	184	**Multicomponent** Intervening work dynamics to reduce sitting time: addressing the individual, environmental and organisational component.	There were no differences between the intervention and control groups regarding the total sitting time after follow-up, nor were there significant changes in health and work behaviours.
Santos et al. [34]	2020	Brazil	204	**Physical Activity** Strength (IG) and generic (CG) exercise programmes to improve musculoskeletal health.	There were no significant changes between IG and CG, although both groups improved in perceived fatigue and muscle strength. After the 4-month follow-up, both groups significantly improved in all outcome variables (perceived fatigue, muscle strength, perception of mental health risk factors, vital signs, and productivity).
Shrivastava et al. [29]	2017	India	267	**Multicomponent** Educational programs focus on physical activity, nutrition, and healthy living.	The intervention group showed a significant decrease in weight, BMI, abdominal circumference, and triglycerides, as well as an increase in HDL levels.
Taylor et al. [35]	2016	USA	69	**Physical Activity** Health promotion programme “Booster breaks”: 15-min breaks during the working day.	The intervention group had better weekly pedometer counts, significantly decreased sedentary behaviour, and increased leisure time physical activity.

^1^ Post-intervention sample.

As seen in the forest plot graph (Figure 4), the total sample was *n* = 8535 (*n* = 5472 in the intervention group and *n* = 3063 in the control group). The overall effect of the two subgroups was significant (−0.12 [−0.22, −0.02], 95% CI; *p* = 0.01); however, the PI obtained was 95% [−0.37, 0.13]. Similar results were obtained for the multicomponent interventions subgroup (−0.14 [−0.24, −0.03], 95% CI; *p* = 0.009), PI at 95% [−0.42, 0.14]. Finally, for physical activity only interventions, the effect size was not significant (−0.09 [−0.39, 0.21], 95% CI, *p* = 0.56); PI 95% [−1.33, 1.15]. The heterogeneity tests were high, with an inconsistency rate greater than 50% in the three analyses: I^2^ = 59% for the overall analysis, I^2^ = 65% for the physical activity subgroup, and I^2^ = 61% for the multicomponent interventions subgroup. These results underline the high variability between interventions. The sample varied in each subgroup.

Multicomponent interventions were the most numerous, with eight articles and very favourable results for the reduction of BMI. Among the activities analysed, the proposal by Jamal et al. [30] with the lifestyle modification program, which includes a schedule of physical activity, nutrition, and psychological accompaniment, was one of the most effective ones in reducing BMI. Interventions with a prolonged implementation and multilevel approach—education, workshops, and organisational changes—[25,26] had a positive trend towards the intervention, although less significant. Other methods are reducing a sedentary lifestyle, combining educational sessions, promoting physical activity [27], and actions at the environmental and organisational level [24]. Almeida et al. [28] included an educational intervention and economic incentives to motivate workers to lose weight. At the same time, Shrivastava et al. [29] used individualised follow-up of the participants to encourage their educational program. Finally, Noori et al. [31] proposed a specific intervention for women focused on behavioural therapy and empowerment, obtaining significant results in reducing the value of BMI.

Conversely, only four of the selected articles have focused on promoting physical activity in the workplace. Among the interventions, we find active breaks during the workday [35], promotion of an increase in the number of daily steps and reduction of a sedentary lifestyle in the workplace [33], a combination of aerobic and strength exercises [32], or only strength and resistance [34]. For these last two interventions, the authors set up areas within the company so that workers could use them throughout the working day.

The Funnel Plot (Figure 5) shows no homogeneous distribution, giving rise to a possible publication bias. Some included studies in the meta-analysis had a small sample size, showing in the graph some dispersion. However, the number of studies included in each subgroup was insufficient for a clear conclusion. Due to the results in the funnel plot for publication bias and the high inconsistency values shown in the forest plot (I^2^ > 50%), the GRADE analysis (Figure 6) indicated low certainty for the individual and overall analyses. Thus, the multi-component approach effectively reduces BMI, especially in the overweight and obese working population. At the same time, controlled physical activity of 15 min/day every day, or 30 min/day at least five days a week, could effectively reduce BMI in the working population.

## 4. Discussion

In this meta-analysis, 12 articles have been included where different interventions to promote health at work aimed at reducing BMI were proposed. The characteristics of the selected interventions have been heterogeneous, making their interpretation complex.

Although the results were not significant, the subgroup of interventions promoting physical activity has shown a positive trend in reducing BMI. Among the different types of physical activity, those carried out during working hours and controlled are the ones that have obtained the best values. However, there needs to be greater clarity in the scientific literature regarding the choice of PA during working hours versus leisure time. Activities performed during working hours allow the worker to take breaks, relieving symptoms of stress and anxiety [36]. It also improves a sedentary lifestyle during the working day [10] and prevents musculoskeletal problems derived from the workplace [37]. Blafoss et al. [38] demonstrated that older people with physically demanding jobs spent fewer hours doing PA during their leisure time, negatively affecting their productivity. These authors highlighted the need to implement PA programs during working hours to improve physical activity and help these workers meet the demands of their job. Similar results were found in Cook and Gazmarian’s [39] study on overweight workers. However, other authors emphasize the importance of promoting PA during free and leisure time, relying on the benefits for the well-being of workers, particularly in the psychosocial domain, reducing stress levels and helping disconnect from work [40,41,42].

The second subgroup, directed at multicomponent interventions, has been shown to obtain significant results in reducing workers’ BMI. These interventions, which promote PA and healthy dietary habits, psychological support, and environmental changes in the company, are essential for achieving and maintaining behavioural changes over time [43,44]. In addition, in those jobs where workers must spend most of their time doing a sedentary activity, such as office work, taking short breaks throughout the workday and exercising the muscles is essential [45]. However, some barriers can limit the performance of these interventions, such as sweating [46] or feeling that too much time is wasted if more than two active breaks are taken during the day [47]. Regarding the nutritional aspect, a common focus of these interventions is teaching workers to choose healthy dishes and foods both while working and during their free time. This is crucial, mainly so that workers can learn how to include healthy foods in their daily diet [48]. However, studies have underlined the difficulty of maintaining a healthy diet due to the increased cost of certain products in some countries, making healthy foods unaffordable for workers with lower incomes [49,50]. One potential solution could be to offer healthy dishes at reasonable prices in the workplace [51].

It is essential for workers to receive psychological coaching to adopt appropriate healthy behaviours and address potential barriers [52]. In this regard, one proven effective tool for maintaining fluid communication is to use technology, such as sending emails, social media, or phone messages [53]. Furthermore, teamwork can motivate lifestyle changes, using peer support to encourage participation in WHP programs [54]. All this should be included in the company policy, where changes at the organisational level would allow these activities to be integrated into each type of work [55,56]. The analysis results support this perspective, where the multi-component approach is the most comprehensive option to tackle the problem of overweight and obesity. In addition, some authors are beginning to question whether the paradigm followed for health promotion in public health, where the target population is general and not individual, is truly useful for WHP [57]. This paradigm focuses on the recent studies’ tendency to have better results in specific interventions for risk groups than those aimed at a general population of workers [58].

The fact that the assessment of the risk of bias has been medium-low should be highlighted. It is noteworthy that the items that obtained a high risk of bias were related to blinding. In most studies, it was difficult to blind the personnel leading the intervention because they also participated in the study. For participants, in those interventions implemented within the same company workers could share information, being difficult to blind the activities. The degree of certainty of the meta-analysis could have been higher, mainly due to the small number of articles and the high heterogeneity of the results. This presents a significant problem when making general recommendations for the working population. Wilkinson [59] warned in his publication on policies and occupational health about the scarcity of studies evaluating the quality and effectiveness of WHP interventions implemented in companies.

As for the implication for practice, these results are a turning point for all health researchers since they underline the urgent need to evaluate the effectiveness of the interventions. However, this evaluation can only be performed if the interventions begin to be standardised, with agreements reached with the competent national and international organisations in WHP to publish guidelines and protocols that help researchers follow common indications.

The European Network for Workplace Health Promotion (ENWHP) and The National Institute for Occupational Safety and Health of the US are some of the international organisations that provide guidelines and examples of good practices [60,61]. The primary purpose of these organisations is to prioritize WHP as a goal for companies. However, there still needs to be some referents in WHP in some countries where health and social inequities are strongly present.

The interventions included in the analysis were implemented in different countries and cultures. This highlights the issue’s importance, as it affects WHP programs widely. In this way, WHP programs should be evaluated in different populations, which may significantly contribute to research.

Workers’ health should be a global priority and addressed urgently. WHP protects workers from work-related hazards, improves productivity, and reduces economic costs for companies [62]. That makes WHP a growing movement in most countries where companies offer some form of employee health program. However, analyzing the results of these programs can be challenging. To help researchers develop WHP programs, international organisations should consider unifying guidelines. The standardisation of interventions, although difficult, could improve health outcomes in the working population. Future studies should focus on generating new evidence on WHP interventions with high certainty, allowing companies and organisations to incorporate them into their policies.

### Study Limitations

The search strategy, when carried out in a general way for interventions to promote health at work, could have limited the results obtained.

Many protocols and interventions have been found regarding the type of articles that mixed working and non-working populations. Another critical limitation has been the significant heterogeneity of the methodology and instruments for measuring the results and the high degree of blinding and publication bias, making it very difficult to meta-analyse them and make recommendations with acceptable certainty.

Lastly, this systematic review excluded studies written in other languages different from English and Spanish, which could decrease the number of eligible articles.

## 5. Conclusions

The WHP interventions that have had the best results for reducing BMI have consisted of a multicomponent approach compared to only performing physical activity programs, although both approaches obtained positive changes. However, the meta-analysis evaluation showed a low degree of certainty, so caution should be exercised in the recommendations of these WHP programs. These results are a turning point for all health researchers since they underline the urgent need to evaluate the effectiveness of the interventions. WHP programs need to be standardised to carry out quality analyses and to make visible their importance for the well-being of workers and the benefits of the companies and organisations involved.

## Figures and Tables

**Figure 1 healthcare-11-01160-f001:**
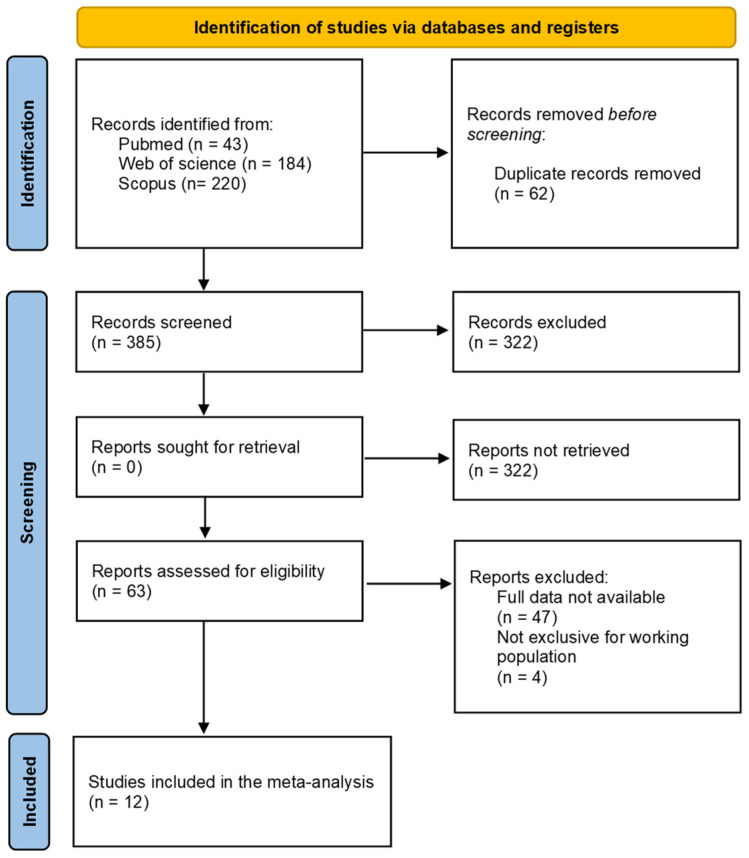
PRISMA flow-chart diagram.

**Figure 2 healthcare-11-01160-f002:**
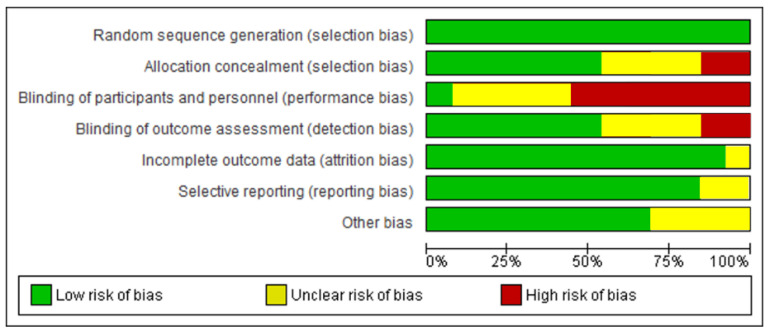
Risk of bias assessment.

**Figure 3 healthcare-11-01160-f003:**
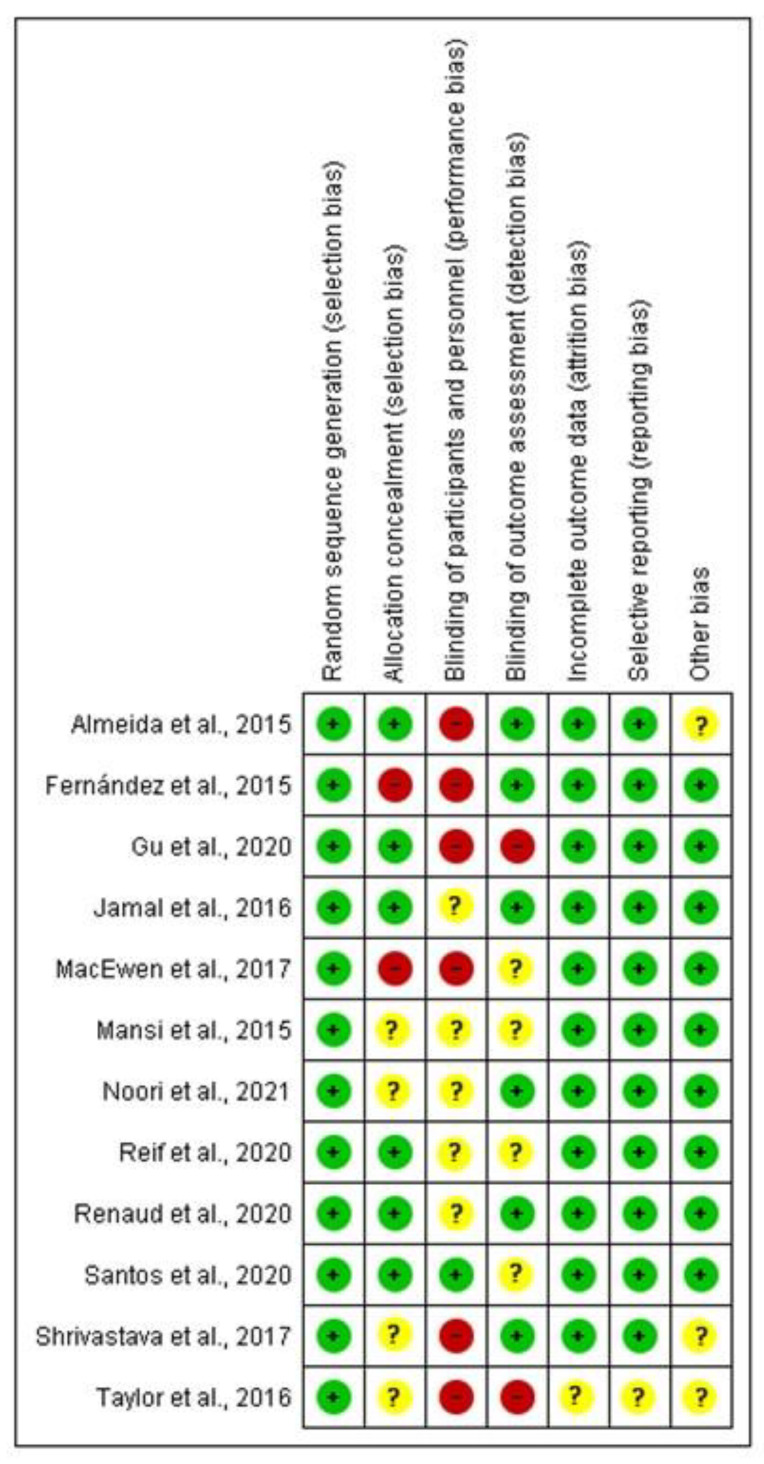
Risk of bias summary. (+) low risk; (−) high risk; (?) uncertain risk [24,25,26,27,28,29,30,31,32,33,34,35].

**Figure 4 healthcare-11-01160-f004:**
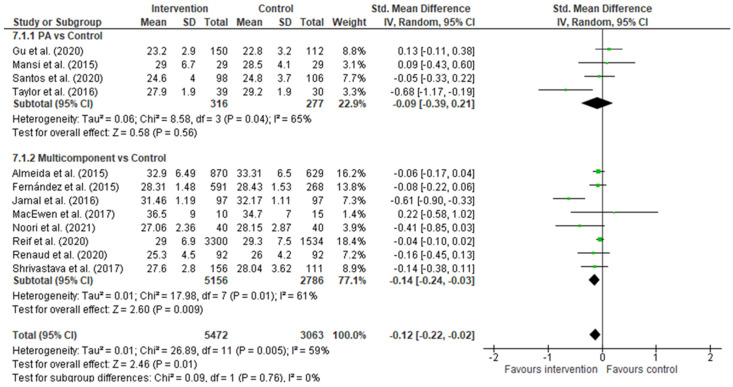
Forest Plot. Black diamond—Overall effect, 95% IC; Green box—study weight [24,25,26,27,28,29,30,31,32,33,34,35].

**Figure 5 healthcare-11-01160-f005:**
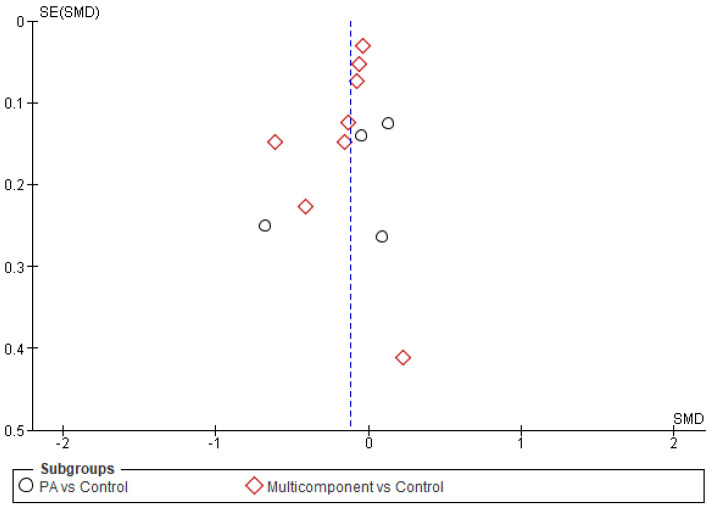
Funnel Plot.

**Figure 6 healthcare-11-01160-f006:**
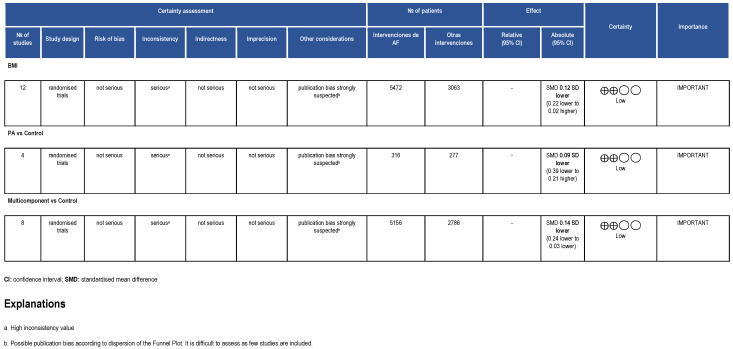
GRADE evaluation.

## Data Availability

No new data were created or analyzed in this study.

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
