# Peer review of "Effectiveness of Multicomponent Interventions and Physical Activity in the Workplace to Reduce Obesity: A Systematic Review and Meta-Analysis"

_healthcare, 2023, doi:10.3390/healthcare11081160_

Round 1

Reviewer 1 Report

This study provides interesting review regarding the effectiveness of multicomponent interventions and physical activity in the workplace to reduce obesity.

Language: The English in the present manuscript requires improvement –in certain passages there are evident inconsistencies regarding grammar/vocabulary and sentence structure. Editing is needed to make sure that the text is at a high standard before publication. Please carefully proof-read to eliminate grammatical errors and improve sentence construction, word choice and clarity.

The title accurately reflects the content and, in general, the abstract presents an adequate synopsis of the paper. The introduction provides a good, generalized topic background with a logically organized and well-argued narrative. Methods and search protocol are clearly articulated and described in sufficient detail. The ambition and conclusions of the paper are fitted with the scope of the journal. Research question is relevant and the conclusions consistent with the evidence and arguments presented. The authors reported on the completion of an electronic search for potentially eligible articles as well as on the performance of any risk of bias assessments of the included studies.

Reviewer 2 Report

This paper makes a contribution to the body of knowledge. The study is designed clearly, has a conventional approach to conducting a meta-analysis, and is executed competently.

The study is ‘set up’ effectively in a clearly argued and succinct introduction. The Methods section is also clear and would enable a replication study to be conducted.

The results are presented in an accessible format and summarise the data and findings of the studies that were reviewed effectively.

The discussion is consistent with the results and the conclusions (though brief) are also coherent.

Specific comments:
4              My preference would be for ‘meta-analysis’ rather than metanalysis. But whichever term is chosen, it needs to be used consistently throughout (e.g., 71 mata-analyse)

22           The comment in the Abstract about ‘low certainty due to high heterogeneity’ needs explanation.

67           The observation that there needs to be more heterogeneity in the implementation of health promotion interventions requires some elaboration. It is asserted here without explanation / rationale. Clarification on ‘study heterogeneity’ is also needed to be able interpret and understand explanation of research design (125) and consideration of the results (153).

88           As an aside, is there any reason to think there may be a significant contribution to this work in languages other than English and Spanish? If so, this might be a recommendation that emerges from the study.

187 & 188            Figure 5 and Table 2 require more explanation to understand the conclusion about ‘no homogeneous distribution’ and ‘possible publication bias’.

There are some inaccuracies in the written English:

14           metanalyses

31           don’t leave preposition (‘from’) at the end of the sentence.

52           Is ‘field’ the right choice of word?

204         Order of subordinate clause means that the sentence doesn’t ‘scan’ well.

231         ‘during and outside’

Other suggestions

15           BMI in full first time

179         Figure 4 will be clearer on a landscape page (like Table 1).

Reviewer 3 Report

Thank you for submitting your work as a potential publication to the journal "Healthcare". Overall, the paper is interesting, an focuses on an aspect of paramount importance. The manuscript is thorough, interesting and the conclusions support and endorse the results obtained. It is an interesting paper. However, some aspects should be improved with the aim of enhancing the general quality of the paper: (1) the manuscript is generally well written, yet some paragraphs and sentences are awkward and/or incorrect (e.g. “This paper aims to metanalyses the effectiveness of workplace health promotion interventions to reduce BMI” or

“As seen in the forest plot graph (figure 4), the total sample was in the intervention group n=5472 and the control group n=3063”). A thorough review by an English native writer/speaker would be highly recommendable; (2) WHP programs need to be standardized, as you clearly highlight, but further information on what efforts should be channeled to standardize and on what particular guidelines should be followed for the aforementioned standardization should be developed, in my opinion. This could be a point of discussion that deserves further development.
